# Analysis of Electric Field Stimulation in Blue Light Stressed 661W Cells

**DOI:** 10.3390/ijms24043433

**Published:** 2023-02-08

**Authors:** Sharanya Bola, Pallavi Subramanian, Daniela Calzia, Andreas Dahl, Isabella Panfoli, Richard H. W. Funk, Cora Roehlecke

**Affiliations:** 1Institute of Anatomy, TU Dresden, D-01304 Dresden, Germany; 2Institute of Clinical Chemistry and Laboratory Medicine, TU Dresden, D-01069 Dresden, Germany; 3Department of Pharmacy—DIFAR, Biochemistry and Physiology Lab., University of Genoa, 16126 Genova, Italy; 4Deep Sequencing Group SFB 655, Biotechnology Center, TU Dresden, D-01069 Dresden, Germany

**Keywords:** electrical stimulation, photoreceptors, blue light, membrane potential, unfolded protein response

## Abstract

Though electrical stimulation is used as a therapeutic approach to treat retinal and spinal injuries, many protective mechanisms at cellular level have not been elucidated. We performed a detailed analysis of cellular events in blue light (Li) stressed 661W cells, which were subjected to direct current electric field (EF) stimulation. Our findings revealed that EF stimulation induced protective effects in 661W cells from Li-induced stress by multiple defense mechanisms, such as increase in mitochondrial activity, gain in mitochondrial potential, increase in superoxide levels, and the activation of unfolded protein response (UPR) pathways, all leading to an enhanced cell viability and decreased DNA damage. Here, our genetic screen results revealed the UPR pathway to be a promising target to ameliorate Li-induced stress by EF stimulation. Thus, our study is important for a knowledgeable transfer of EF stimulation into clinical application.

## 1. Introduction

How can electrical stimulation assist during injuries? Where is the rational coupling of electrical stimulation to cell and molecular biology? Studies show that endogenous electric fields are an important cue in early embryogenesis, wound healing, and regeneration [1,2,3,4,5,6]. Interestingly, multiple studies have shown that electrical stimulation is also known to be effective in protecting a degenerating retina. A variety of approaches, such as subretinal-, transcorneal- or whole eye electrical stimulation, showed a partial recovery of visual function [7,8,9,10,11,12,13,14].

In the present study, we mimicked retinal stress with blue light (Li) as a specific stressor for the retina, and especially for the photoreceptors. The blue light damaging effect (blue light hazard) is gaining general and clinical awareness because the exposition to artificial light as ambient illumination and in screens of electronic devices increases drastically [15,16]. In animal studies, damaging effects were also found in the retinae of different rat strains with cell loss of the pigmented epithelium [17]. Studies in live explant retinae show that Li can induce oxidative stress [18,19], protein oxidation, and lipid oxidation in photoreceptors [20], cellular dysfunction [21], disrupt mitosis [22], promote DNA damage [18], and induce apoptosis [23] and cell death [18,24]. 

Therefore, the aim of this study was to test possible protective mechanisms on retinal photoreceptor cells damaged by blue light. We especially wanted to understand how direct current electric fields (EF) induce these effects at the cellular and molecular level. 

To test the hypothesis, we used EF field strength of 0.25 V/cm for a duration of 10 min to stimulate the cells exposed to Li of 1.5 mW/cm^2^ intensity for 4 h. To look for possible EF-induced cellular responses in Li-stressed 661W cells, we looked at cell viability, metabolism with mitochondrial function, and searched for up-regulated genes and pathways involved in protection. All the data we gathered showed a coherent picture of balancing out the Li-stressed cells regarding ROS production, redox, and mitochondrial respiratory state leading to higher ATP production. An absolute novel finding is that EF stimulation tackles the stress via activated unfolded protein response (UPR) signaling pathway. 

## 2. Results

### 2.1. Effects of EF Stimulation in Li Exposed Cells at Cellular Level

#### 2.1.1. EF Induces Sustained Overall Cell Viability

Cell morphology was studied by microscopy following 4 h blue light (Li) exposure in 661W cells (Appendix A). Li exposed cells were round in shape in contrast to untreated cells exhibiting a normal morphology, with thin membrane protrusions extended in all directions. The additional treatment of Li-exposed cells with EF (LiEF) resulted in a gradual extension of their membranes towards cathode, similar to the cells stimulated with EF alone. Li treatment significantly reduced the percentage of live cells by 10.89%, 21.67%, and 16.97%, at 0 h, 3 h, and 6 h, respectively, compared to control cells (Figure 1A–D). Interestingly, stimulation with EF following Li exposure significantly increased live cell percentage by 1.26%, 13.68%, and 10.84%, at 0 h, 3 h, and 6 h, respectively, compared to Li cells. Furthermore, immunofluorescence results of phosphorylated histone H2AX (γH2AX) for DNA damage indicated no difference in control and EF stimulated cells (Appendix A). However, LiEF cells exhibited less γH2AX foci per cell compared to Li exposed cells (Appendix A). In addition, cell cycle analysis revealed no significant increase in the percentage of cells in S and G2/M phase between the Li and LiEF cells at 3 h and 6 h time points (Appendix A). This suggests that reduced DNA damage in LiEF- treated cells was not due to enhanced cell proliferation but by the inhibition of Li induced cell death and DNA damage due to EF stimulation.

#### 2.1.2. EF Depolarizes the Plasma Membrane Potential

Studies showed that cell survival, proliferation, and migration are related to the plasma membrane potential (V_m_) [25]. We analyzed the V_m_ of cells using anionic fluorescent dye DiBAC_4_ (3) by fluorescence microscopy and flow cytometry at 0 h time point (Figure 2). In control cells, we observed the random accumulation of dye. In contrast, we noticed a relatively hyperpolarized state of the cell membrane on the leading edge, compared to depolarized rear end in EF stimulated cells. However, Li exposure hyperpolarized the plasma membrane (Figure 2A,C). The fluorescence intensity profile of membrane potential measured at flow cytometry revealed 24% decrease in Li cells and 25% increase in EF cells compared to control cells (Figure 2B). Moreover, 35% increase in intensity was observed in LiEF cells compared to Li cells (Figure 2C,D). 

#### 2.1.3. EF Decreases Oxidative Stress 

It is known that Li induces oxidative stress in retinal cells [26,27]. We studied whether EF further induces or prevents oxidative stress in LiEF-treated cells by measuring intracellular levels of heme-oxygenase 1 (HO-1) and superoxide. Gene levels of HO-1 were significantly raised by 12.73-fold after Li exposure as compared to control cells at 3 h as measured by qRT-PCR (Appendix A). However, LiEF treatment significantly reduced this effect by 0.61-fold compared to Li cells. Western analysis revealed a significant increase in HO-1 protein levels in Li cells compared to control cells at 3 h and 6 h. Additionally, in LiEF cells we observed a significant increase in HO-1 protein levels by 13.82-fold compared to control cells. Of note, decrease in HO-1 protein levels in LiEF cells was seen at 6 h compared to Li cells (Appendix A). 

Moreover, studies have shown that oxidative stress is known to increase ROS levels [28]. Here, we examined the ROS levels by staining with dihydroethidium (DHE), which detects superoxide radicals using microscopy and flow cytometry (Appendix A). As shown by flow cytometry at 3 h time point, significant increase in DHE fluorescence by 32% and 55% was observed in Li exposed and LiEF-treated cells respectively compared to the control cells (Appendix A). Additionally, a significant increase in DHE fluorescence by 24% was seen in LiEF-treated cells compared to Li-exposed cells at 6 h. Likewise, the protein levels of NADPH 4 oxidase as a major source of ROS were increased by 0.39-fold and 0.27-fold in Li and LiEF cells, respectively, at 3 h compared to control cells (Appendix A). Furthermore, 0.50-fold and 0.55-fold increase of NADPH 4 oxidase in Li and LiEF cells with respect to control cells was observed at 6 h.

#### 2.1.4. EF Enhances Cell Migration

As migration is widely studied process during electrotaxis [3], we studied migration by time lapse microscopy. Li exposure impaired the accumulated migration distance and velocity of cells by 0.49 and 0.58-fold, respectively, compared to control cells (Appendix A). Interestingly, LiEF treatment significantly increased the accumulated distance and velocity by 0.77 and 0.42-fold compared to Li treatment, respectively. Additionally, EF stimulation enhanced accumulated distance and velocity.

### 2.2. Effects of EF on Mitochondria and Its Activity in Li Exposed Cells

#### 2.2.1. EF Promotes Gain in Mitochondrial Membrane Potential

We studied EF effect on mitochondrial membrane potential (ΔΨ_m_) by fluorescence microscopy and flow cytometry using JC- 1 dye at 0 h (Figure 3A,B). Our data show a significant decrease in ΔΨ_m_ in Li cells compared to control and LiEF cells. In contrast, increased ΔΨ_m_ was observed in LiEF compared to Li cells (Figure 3B). Additionally, mitochondrial (mt) ROS levels measured using MitoSOX dye by fluorescence microscopy showed a significant increase in mtROS by 3.1 and 2.9-fold at 3 h and 6 h, respectively, in Li cells compared to control cells (Appendix A). Decreased mtROS was seen in LiEF cells compared to control cells. Notably, mtROS in EF cells remained unaltered compared to control cells. The increase in mtROS in Li exposed cells compared to control cells might be due to the loss of ΔΨ_m_ observed at 0 h. Furthermore, we analyzed mitochondrial respiration and ATP synthesis in the presence of malate and pyruvate at 0 h (Figure 3C,D). ATP production was inhibited by 30% in Li cells compared to control cells. On contrary, increased ATP production was seen in LiEF cells compared to Li cells. Likewise, ADP stimulated oxygen consumption was impaired in Li cells. Of note, respiratory capacity was restored in LiEF cells compared to Li cells. Our results show a restored mitochondrial function in LiEF cells with an increase in ATP production and oxygen consumption. 

#### 2.2.2. EF Increases Mitochondrial Respiratory Capacity

We examined the mitochondrial morphology using MitoTracker dye by microscopy. We found a significant increase in mitochondrial content and length of mitochondria in LiEF cells compared to Li cells (Appendix A). Given the observation about oxygen consumption and ATP production (Figure 3C,D), we analyzed mitochondrial respiratory capacity using extracellular flux analysis. Oxygen consumption rate (OCR) reflecting the rate of mitochondrial respiration was increased in LiEF cells compared to Li exposed cells (Figure 4A,B). Additionally, a significant increase in basal respiration by 0.92-fold (Figure 4C), maximal respiration by 0.96-fold (Figure 4D), and ATP production by 0.92-fold (Figure 4E) was observed in LiEF compared to Li cells. Consistently, spare respiratory capacity was significantly reduced in Li cells compared to control cells and this was restored to normal levels in LiEF- cells (Figure 4F). Furthermore, the protein levels of respiratory chain complexes I, II and III were increased in LiEF cells compared to Li cells (Appendix A) with no change in protein levels of complexes IV and V (Appendix A). Our data might suggest that an increased mitochondrial content could accommodate increased ATP requirements with highly efficient respiratory chain complexes. The relevance of these findings indicate that EF stimulation restores the mitochondrial respiratory capacity in LiEF treated cells.

### 2.3. Effects of EF on Li Exposed Cells at Signaling Level

#### EF Upregulates Unfolded Protein Response (UPR) Pathway

Our RNA-sequencing analysis revealed a total of 5067 and 2958 genes differentially regulated in Li and LiEF cells with respect to control. We included the heat map of top 25 upregulated and downregulated genes from Li and LiEF with respective to control cells (Figure 5A,B). Ingenuity pathway analysis showed 19 significantly affected pathways with an upregulated unfolded protein response (UPR) and Ataxia-telangiectasia mutated (ATM) signaling pathways in LiEF cells suggesting a role of EF stimulation to abrogate Li-induced stress. (Figure 5C). Gene Ontology analysis revealed significantly enhanced processes in Li and LiEF cells (Figure 5D). To determine the UPR upregulation in Li and LiEF, we examined three distinct branches of UPR: ATF6, PERK and IRE1α using qRT-PCR and Western blotting (Figure 6A–C). These branches of UPR operate in parallel to regulate the expression of numerous genes in order to maintain cellular homeostasis or induce apoptosis if ER stress cannot be alleviated [29]. Figure 6D shows differentially expressed genes involved in the UPR pathway in Li and LiEF cells as obtained from RNA sequencing data. 

An upregulated PERK branch was observed in Li cells (Appendix A) with a significant increase in gene expression of PERK by 1.80-fold (Appendix A), DDIT3 by 3.50-fold (Appendix A), ATF4 by 1.65-fold (Appendix A) and GADD34 by 1.76-fold (Appendix A) compared to control cells. In contrast, a significant decrease was seen in LiEF cells compared to Li cells (Appendix A). Furthermore, significant decrease in eIF2α was seen in LiEF, compared to Li cells (Appendix A). Interestingly, a significant increase in phosphorylated eIF2α (p-eIF2α) was observed in LiEF compared to Li and control cells which may halt further gene translation to reduce ER load (Appendix A). In consistence with qRT-PCR results, we observed decreased CHOP expression in LiEF cells suggesting the role of EF to suppress apoptosis through down regulation of CHOP (Appendix A).

Protein levels of Hsp70, IRE1α and calnexin were increased in Li and LiEF cells (Appendix A). Moreover, we noticed a significant increase in HSP5A gene levels by 2.64-fold in Li and 2.61-fold in LiEF cells compared to control cells indicating the activation of UPR (Appendix A). Regarding IRE1 branch, a significant 2-fold increase in gene expression of XBP1 was seen in LiEF compared to control cells (Appendix A). Also, an increased protein level of IRE1α was observed in Li and LiEF compared to control cells (Appendix A). Furthermore, immunoblot analyses revealed an increase in chaperone Hsp70 protein levels (Appendix A) and Calnexin (Appendix A) in LiEF compared to Li cells.

We noticed a significant increase in gene expression of ATF6 (Appendix A), along with HERPUD1 (Appendix A) and SYVN1 (Appendix A) in Li and LiEF compared to control cells from ATF6 branch. In addition, protein levels of oxidoreductases were investigated (Appendix A). LiEF stimulation significantly increased the expression of ERO1-Lα compared to Li cells (Appendix A). Additionally, we noticed an increased expression of PDI in Li and LiEF compared to control cells (Appendix A). Here, our results demonstrate that EF stimulation upregulated the UPR pathway to reduce Li-induced stress in LiEF cells.

## 3. Discussion

Considerable research efforts have focused on finding the cellular and molecular events underlying the rescue effect of electrical stimulation in degenerating retinae. Reported effects are that electrical stimulation directs retinal cell axon growth in vitro [30] improves survival of transected retinal ganglion cells (RGC) in rats [31,32], delays degeneration and improves the survival and function of photoreceptors [33], and exhibits neuroprotective effects in the retinas of rats [34,35,36]. Furthermore, the downregulation of pro-inflammatory cytokines like TNF-α, IL-1β and the pro-apoptotic gene Bax [33,37,38] and the expression of neurotrophic factors like BDNF [39], CNTF [35], FGF-2 [32,38], and IGF-1 [31,36], has been studied. However, the molecular mechanisms behind electric field-induced protective effects are mainly unknown. The above mentioned studies focused on the role of neurotrophic factors and cytokines. However, investigating protective effects at the single-cell level to prevent the influence of uncertain factors in tissue is recent trend. 

Firstly, we investigated the immediate effects such as viability, oxidative damage, changes in potential, DNA damage and cell migration. Secondly, effects on bioenergetics were determined by measuring mitochondrial activity also with single cell analysis. Thirdly, we studied UPR pathway as it appeared to be significantly involved in mitigating Li-induced stress as observed by genetic screen tests. By this panel of combined findings, we could tackle which beneficial effects could exert a brief EF stimulation. 

For the first time, we could show the following rescue effects of EF: the rescue of Li-induced changes in cell morphology; the improvement of Li-reduced migration velocity and displacement. Recent evidence supports our data about restored migratory responses in LiEF cells that directional sensing mechanism during electrotaxis is due to a change in resting plasma membrane potential (V_m_) [40]. We also observed the depolarization of V_m_ in LiEF cells as a result of EF stimulation. Studies reported that EF promotes neural stem cell differentiation and neurite outgrowth [41], and inhibits secondary apoptosis [42]. This supports our data of more viable cells in LiEF cells than in Li cells. An increased cell survival in LiEF cells by cell proliferation could be ruled out by our cell cycle analysis. 

Our results show a persistent increase in cellular superoxide levels in LiEF cells, too. This increase could either be due to increased NADPH 4 oxidase levels, which contribute to cell survival [43] or reflecting a possible role of ROS during electrotaxis [44]. In line with previous studies, the formation of γH2AX foci during oxidative stress is considered as an early event upon DNA damage [45]. Here, our immunofluorescence results demonstrate reduced γH2AX foci in LiEF cells compared to Li cells at 6 h, indicating an activation of DNA repair mechanism. Consistently, sequencing data showed a remarkable upregulation of ATM (ataxia-telangiectasia mutated) signaling pathway in LiEF cells. Observed higher HO-1 levels at 3 h compared to 6 h in LiEF reveals the need for high levels of HO-1 at 3 h to maintain cellular homeostasis as demonstrated by Gozzelino et al. [46]. 

Interestingly, we observed short and round mitochondria with a decrease in connectivity in Li cells but an increased mitochondrial connectivity with elongated mitochondria in LiEF cells. Evidence for light affecting mitochondria was demonstrated in cultures of fibroblasts [47] and RGC [48,49]. Our results confirmed that changes in mitochondrial membrane potential (ΔΨ_m_) were dependent on mitochondrial morphology. Interestingly, we observed a significant gain of ΔΨ_m_ in LiEF cells compared to Li cells as a consequence of EF stimulation. This substantiates the effect of EF stimulation on mitochondria, which is supported by our findings regarding mitochondrial morphology, mtROS and activity. We cannot exclude that the gain or loss of ΔΨ_m_ we observed in treated 661W cells may be targeted only to a relatively small area of the mitochondrion compared to the network over the length of tens of micrometres [50,51]. 

Nevertheless, we found enhanced ATP production and oxygen consumption along with decreased mtROS in LiEF cells. This elevated mitochondrial activity in LiEF cells compared to Li cells can be accounted to an increase in OXPHOS protein levels. It is shown here for the first time that EF stimulation rescued impaired mitochondrial function caused by light [52,53]. 

Very few studies demonstrated that bioelectric activities control signaling pathways apart from maintaining cellular homeostasis and behavior [3,54,55,56]. To obtain further insights into the protective role of EF stimulation, mRNA sequencing was performed to observe the triggered signaling pathways along with the various regulated genes. Our results reveal that EF stimulation led to an overall survival of LiEF cells and support the fact that endoplasmic reticulum (ER) stress activation is due to Li exposure as observed during light-induced photoreceptor degeneration [57,58]. It is known that altered ER homeostasis can trigger the unfolded protein response (UPR) pathway to enhance protein folding and assembly [59]. Here, our sequencing results show especially an upregulated UPR pathway, which combats ER stress and regulates ER homeostasis via PERK, IRE1, and ATF6 branches. EF stimulation is also known to recruit signaling molecules to promote cell survival in RGC [60] and light injured photoreceptors [36]. Our results further revealed an enhanced protein expression of phosphorylated eIF2α in LiEF cells indicating an arrest in global protein translation to reduce ER stress [61]. Additionally, an increase in chaperone levels in LiEF cells might assist to suppress ER stress. Furthermore, EF stimulation increased gene expression levels of HERPUD1 and SYVN1 in LiEF cells, which might exert protective effects against ER stress as observed in previous studies [62,63]. Further investigations will be necessary to show how EF stimulation directly triggers the UPR pathway. 

## 4. Materials and Methods

### 4.1. Cell Culture 

Mouse photoreceptor-derived 661W cells were obtained from Muayyad R. Al-Ubaidi (University of Oklahoma, Oklahoma City, OK, USA). Cells were maintained in DMEM supplemented with 10% FCS at 37 °C in a humidified 5% CO_2_ and were passaged every 3 to 4 days. For experimental procedures, 200 μL of cell suspension (10% FCS in DMEM) containing 5000 cells was gently pipetted into channels of µ-Slide I (Ibidi, Martinsried, Germany) and were incubated for 3 h before adding DMEM medium (with 10% FCS). The slides were then incubated at 37 °C with 5% CO_2_ for 48 h before performing any experiment. Around 3–6 slides per sample were seeded with cells to obtain a greater number of cells for flow cytometry, sequencing, qRT-PCR, bioenergetics and western blotting experiments. For every experiment different groups of cells as blue light irradiated (Li; 1.5 mW/cm^2^, 405 nm, 4 h), EF stimulated (EF; 2.5 V/cm, 10 min), EF stimulation of irradiated cells (LiEF; 1.5 mW/cm^2^, 405 nm, 4 h; 2.5 V/cm, 10 min) and no EF-no light (control) cells were used. Post exposure to blue light and EF stimulation, the slides were incubated for indicated hours as mentioned in Appendix A for analyses. We have chosen 6 h maximum because the Ibidi slides were seeded with cells and incubated for 48 h prior to conducting any experiment. At this point, the experiments started. Longer periods (more than 12–24 h) would lead to confluent slides, not suited for carrying out the experiments. 

### 4.2. Experimental Setup

Cells were stimulated with direct current electric fields (EF) using a model as described previously [64]. During our study, EF was applied to cells through constant voltage power supply via two platinum electrodes (0.2 mm diameter) immersed in glass bottles filled with 0.9% NaCl. These electrodes were submerged in brine solution, connecting the power supply and agar bridges. The slides used were smaller in dimensions in order to decrease the surface area and increase the field strength to the cells or tissues. An amperometer was used to measure the EF at the beginning of experiment to ensure that right field strength was applied to cells. In this study, field strength of 2.5 V/cm for 10 min was applied at 37 °C, 5% CO_2_. In order to monitor temperature increase, a thermocouple was used to monitor changes if any. An Olympus IX81 inverted microscope equipped with a motorized stage, incubation system (Carl Zeiss, Jena, Germany) and xcellence software was used to monitor cell migration. Irradiation was done by a LED based system (# LZ1-00UA05 BIN U8; LedEngin, San Francisco, CA, USA) constructed in our lab which radiates blue light of wavelength 405 nm. The LEDs were arranged in a 2 × 3 grid pattern. All the irradiation and EF stimulation steps were carried out at 37 °C, 5% CO_2_. 

### 4.3. Quantitative Analysis of Electrotaxis 

Electrotaxis was analyzed by using chemotaxis plugin in ImageJ along with Chemotaxis and migration software (Ibidi, Martinsried, Germany). All adherent cells in the images were tracked at 30 min. frame intervals. The position of a cell was defined by its centroids. Moreover, the cell migration rate was quantified as the track speed which was presented as the accumulated migration distance per 30 min. At least 40 cells in triplicates from three independent experiments were analyzed.

### 4.4. Antibodies 

Primary antibodies against PDI (rabbit monoclonal; 1:1000), Ero1-Lα (rabbit monoclonal; 1:1000), IRE1α (rabbit monoclonal; 1:1000), Calnexin (rabbit monoclonal; 1:600), eIF2α (rabbit polyclonal; 1:800), P-eIF2α (rabbit monoclonal; 1:600), and CHOP (mouse monoclonal; 1:800) were purchased from Cell Signaling Technology, Frankfurt am Main, Germany. Antibodies NADPH 4 oxidase (rabbit polyclonal; 1:800), and Total OXPHOS cocktail antibody (rodent monoclonal; 1:800) were purchased from Abcam, Cambridge, UK. Hsp70 (mouse monoclonal; 1:800), and HO-1 (mouse monoclonal; 1:800) were purchased from Enzo life sciences, Lörrach, Germany. β-actin (rabbit polyclonal; 1:1200) from Novusbio, Centennial, CO, USA, and Vinculin (mouse monoclonal, 1:500) from AbD Serotec, Puchheim, Germany. Where applicable secondary antibodies used during immunoblotting were Donkey anti rabbit (1:8000; Novex), Horse anti mouse (1:2000; Cell Signaling Technology), Donkey anti mouse (1:5000; Bethyl), Goat anti rabbit (1:2000; Cell Signaling Technology), and Donkey anti rabbit (1:4000; Novex).

### 4.5. Live/Dead Assay

Intracellular esterase activity and the loss of plasma membrane integrity in cells were measured by staining with Calcein AM and ethidium homodimer-1 (EthD-1) respectively using LIVE/DEAD^®^ Viability/Cytotoxicity Kit (Invitrogen, Darmstadt, Germany). 0.5 mL of PBS containing 0.5 μM of Calcein AM and 6 μM of ethidium homodimer-1 (EthD-1) was added to each sample and incubated at 37 °C with 5% CO_2_ for 15 min. The staining solution was removed, and the samples were washed gently with 2 mL of PBS. Later 2 mL of HEPES buffer was added slowly into the channels of slides and were immediately imaged under Olympus IX81 inverted fluorescence microscope with 494 nm (green, Calcein) and 528 nm (red, EthD-1) excitation filters. Images were captured using xcellence software. For quantitative analysis, over 6–7 areas were randomly chosen in each slide and 150 cells in triplicates were counted using cell counter plugin in ImageJ.

### 4.6. Membrane Potential (V_m_) Measurements

DiBAC_4_ (3) (Molecular Probes, Eugene, OR, USA) was used to measure plasma membrane potential by flow cytometry and microscopy. 0.5 mL of 0.5 μM (in DMEM with 10% FCS) dye was added into the slides and incubated for 30 min in dark at 37 °C. Cells were gently dislodged from surface using 1 mL of non-enzymatic cell dissociation solution (Sigma Aldrich, Schnelldorf, Germany) and around 10,000 cells were added into flow cytometry tubes. Later, cells were centrifuged and suspended in 0.25 mL of PBS and immediately analyzed on flow cytometer at 488 nm. 5000 cells in triplicates per sample were counted. An aliquot of cell suspension was separated to register baseline value. The sampling interval of DiBAC_4_ (3) fluorescence measurements were in the range of 4 and 5 s. Data analysis was performed using CellQuest software (Becton Dickinson, Heidelberg, Germany). 

For microscopy, cells were incubated with 0.5 mL of 0.1 μM DiBAC_4_ (3) dye (in DMEM with 10% FCS) for 5 min at 37 °C. Cells were washed once with 1 mL PBS and immediately imaged under Olympus IX81 inverted fluorescence microscope with excitation maxima of 490 nm. Cells were imaged at 6–8 random places from the slides. Mean fluorescence intensity of 40 cells in triplicates was analyzed using ImageJ. 

### 4.7. Mitochondrial Membrane Potential (Δψ_m_) Measurements

Mitochondrial membrane potential was measured using JC-1 (Molecular Probes, Eugene, OR, USA). Cells were gently dislodged using 1 mL of non-enzymatic cell dissociation solution (Sigma Aldrich, Schnelldorf, Germany). 5 µg/mL of JC-1 dye (in DMEM) was added to flow cytometry tubes containing 10,000 cells each and incubated for 30 min at 37 °C in dark. Cells were centrifuged and immediately measured on flow cytometer with healthy cells as JC-1 aggregates (excitation/emission = 540/605 nm) and apoptotic or unhealthy cells as JC-1 monomers (excitation/emission = 480/510 nm). At least 10,000 cells in triplicates per sample were acquired during each experiment. Data analysis for mean fluorescence intensity was obtained using Cell Quest software (Becton Dickinson, Heidelberg, Germany). Results presented as a change in ratio of two fluorescence means correlates to changes in potentials of mitochondria. For microscopy, cells were incubated with 2.5 µg/mL of JC-1 dye (in DMEM) for 30 min at 37 °C. Later, cells were washed with 1 mL PBS and immediately imaged for green (480 nm) and red (540 nm) fluorescence at Olympus IX81 inverted microscope. 

### 4.8. Superoxide Measurement

DHE is a reduced form of widely used DNA dye ethidium bromide. Upon reaction with superoxide anion, DHE forms a red fluorescent product, 2-hydroxyethidium with excitation and emission at 500 nm and 580 nm, respectively. Cells were incubated with 0.5 mL of DMEM containing 10 μM dihydroethidium (DHE; Invitrogen, Carlsbad, CA, USA) for 30 min in dark at 37 °C. Once dislodged using 1 mL of non-enzymatic cell dissociation solution (Sigma Aldrich, Schnelldorf, Germany), cells were centrifuged and assayed immediately by flow cytometry for superoxide production, indicated by an increase in FL-2 fluorescence (excitation/emission at 518 nm/605 nm). Around 10,000 events in triplicates per sample were counted and mean fluorescence intensity was analyzed using Cell Quest software (Becton Dickinson, Heidelberg, Germany). For microscopy, cells were incubated with 0.5 mL of DMEM containing 5 μM DHE and incubated for 15 min at 37 °C in dark. Once washed with 1 mL of PBS, cells were imaged at 6–8 random places under Olympus IX81 inverted microscope.

### 4.9. Mitochondrial ROS Measurement

For detecting mitochondrial ROS and morphology, cells were incubated with 0.5 mL of HEPES medium containing 0.5 µM Mitotracker Green (Molecular Probes, Eugene, OR, USA) and 1.5 µM Mitosox Red (Molecular Probes, Eugene, OR, USA) for 20 min at 37 °C. Cells were washed thrice with 1 mL of warm HEPES buffer. Once washed, cells were imaged immediately at 6–8 random places under Axio Observer Z1Apotome (Zeiss, Jena, Germany) equipped with an incubation system to maintain 5% CO_2_ and 37 °C at 490/516 nm (green) and 510/580 nm (red). Networks of mitochondria were analyzed using Mitochondrial Network Analysis (MiNA) toolset macros used along with ImageJ [65]. Mean fluorescence intensity of mitochondrial ROS was calculated using ImageJ. Around 40–50 cells were counted for analysis. 

### 4.10. Oxygen Consumption Measurements

An amperometric electrode (Unisense-Micros respiration, Unisense A/S, Aarhus, Denmark) was used to measure the oxygen consumption. The experiment was performed in a closed chamber at 23 °C. For each experiment, around 2 × 10^5^ cells (0.04 mg) were permeabilized with 0.03 mg/mL digitonin for 1 min, centrifuged for 9 min at 1000 rpm and resuspended in: 137 mM NaCl, 0.7 mM NaH_2_PO_4_, 5 mM KCl, and 25 mM Tris HCl (pH 7.4). The same medium was used in the oximetric experiments. Ten mM pyruvate and 5 mM malate were added to the sample to stimulate OXPHOS machinery. To study if OXPHOS is coupled, 0.1 mM ADP was added 2 min after oxidative substrate addition.

### 4.11. ATP Synthesis Assay

ATP formation from ADP and inorganic phosphate (Pi) in 661W cells was measured by the luciferin/luciferase chemiluminescent method (Roche Applied Science, Penzberg, Germany), as described previously [66]. Cells (5 µg protein) were permeabilized with 0.03 mg/mL digitonin for 1 min, centrifuged for 9 min at 1000 rpm and resuspended in 50 mM Tris HCl (pH 7.4), 5 mM KCl, 1 mM EGTA, 5 mM MgCl_2_, 0.6 mM ouabain, 5 mM KH_2_PO_4_, 5 mM pyruvate and 2.5 mM malate and ampicillin (25 µg/mL). ATP synthesis was induced by adding 0.3 mM ADP. 

### 4.12. Mitochondrial Bioenergetics

Seahorse XFe96 Extracellular Flux Analyzer and XF Cell Mito Stress Test Kit were used to measure oxygen consumption rate (OCR) according to manufacturer’s protocol (Agilent Technologies, Santa Clara, CA, USA). After treatment, cells from slides were dislodged using 1 mL of non-enzymatic cell dissociation solution and plated in to custom XFe96 polystyrene well plate (Seahorse Biosciences, North Billerica, MA, USA) at a seeding density of 1 × 10^5^ cells/well in 6–10 replicates and left undisturbed for 3 h. Prior to time of assay, cells were washed with pre-warmed growth medium (100 mL RPMI medium supplemented with 10 mM glucose, 1 mM sodium pyruvate with pH adjusted to 7.4 and filter sterilized) and cell plates were incubated in a non CO_2_ incubator at 37 °C for 30 min to allow pre-equilibration with assay medium. During assay, cells were treated sequentially with 2 µM Oligomycin, 2 µM FCCP, and 0.5 µM Rotenone/Antimycin A. Wave software was used to design, run, and collect the results. The DNA content in each well was determined using CyQUANT cell proliferation assay kit (Molecular Probes, Eugene, OR, USA) to normalize OCR data. The other parameters such as basal respiration, maximal respiration, ATP production and spare respiratory capacity were quantified based on OCR data obtained (refer Figure 6A) [67]. 

### 4.13. Immunofluorescence

Cells were rinsed briefly for 1 min with 2 mL PBS and fixed with 4% PFA for 5 min at room temperature. Once fixed, cells were gently washed with 2 mL PBS and permeabilized with 0.5 mL of 100% ice cold methanol for 10 min at −20 °C. Once rinsed with 2 mL of PBS for 1 min, cells were blocked with 2% BSA/PBS for 30 min at room temperature. Subsequently, 0.25 mL of rabbit polyclonal Phospho-Histone H2AX (1:150; Cell Signaling, Danvers, MA, USA) primary antibody was added to the cells and incubated for overnight at 4 °C. Following day cells were washed thrice with 2 mL PBS for 5 min each and incubated with 0.25 mL of goat anti rabbit Alexa Fluor 488 (1:800, Abcam, Cambridge, UK) secondary antibody at room temperature for 1 h in the dark. Cells were rinsed thrice with 2 mL PBS for 5 min each and incubated with DAPI (1:2500, Sigma Aldrich, Schnelldorf, Germany) for 10 min. Finally, cells were mounted with 2.5% DABCO (in PBS/Glycerol) solution. Around 200 cells were imaged at 6–8 random areas of the µ-slide using wide field microscope Axio Observer Z1apotome (Zeiss, Jena, Germany) controlled by Zen software (Zeiss, Jena, Germany). 

### 4.14. Cell Cycle Analysis

Post incubation, cells were briefly rinsed with PBS. 10 µM of labeling EdU solution was added to the cells and incubated at 37 °C for 1 h. Labeling solution was removed, and cells were fixed and permeabilized according to manufacturer’s instructions (Click-iT EdU, Base Click, Neuried, Germany). Once done, 5000 cells in triplicates were measured using flow cytometer. 

### 4.15. RNA Isolation and Sequencing

For RNA isolation, 3 h post treatment, cells in µ-Slide I were trypsinized and centrifuged at 150× *g* for 5 min. Supernatants were aspirated and RNA was isolated from 5 × 10^5^ (6–8 µ-slides) cells according to manufacturer’s instructions (RNeasy plus mini kit, Qiagen, Hilden, Germany). RNA integrity was checked using 2100 bioanalyzer before subjecting to sequencing. mRNA was isolated from 1 µg total RNA by poly-dT enrichment using the NEBNext Poly(A) mRNA Magnetic Isolation Module according to the manufacturer’s instructions. After chemical fragmentation the samples were directly subjected to strand specific RNA-Seq library preparation (Ultra Directional RNA Library Prep, NEB). For adapter ligation custom adaptors were used (Adaptor-Oligo 1: 5′-ACA-CTC-TTT-CCC-TAC-ACG-ACG-CTC-TTC-CGA-TCT-3′, Adaptor-Oligo 2: 5′-P-GAT-CGG-AAG-AGC-ACA-CGT-CTG-AAC-TCC-AGT-CAC-3′). After ligaton adapters were depleted by SpriBead bead purification (Beckman Coulter, Brea, CA, USA). Sample indexing was done in the following PCR enrichment (15 cycles). For Illumina flowcell production, samples were equimolarly pooled and distributed on all lanes used for 75 bp single end sequencing on Illumina HiSeq 2500.

### 4.16. Bioinformatics Analysis

Alignment of the short reads to the mm10 transcriptome was performed with GSNAP [68] and a table of readcounts per gene was created based on the overlap of the uniquely mapped reads with the Ensembl Genes annotation v. 81 (July 2015) for mm10, using featureCounts (v. 1.4.6) [69]. Normalization of the raw readcounts based on the library size and testing for differential expression between the different cell treatments was performed with the DESeq2 R package (v.1.8.1) [70]. Sample to sample Euclidean distance as well as Pearson’s correlation coefficient (*r*) were computed based on the normalized gene expression level in order to explore correlation between biological replicates and different libraries. For testing for differential expression, the count data were fitted to the negative binomial distribution and the *p*-values for the statistical significance of the fold change were adjusted for multiple testing with the Benjamini-Hochberg correction for controlling the false discovery rate [71]. Accepting a maximum of 10% FDR resulted in 5067 (Control to Light) and 2958 (Control to LiEF) differentially expressed specific genes. 

### 4.17. Differential Genes Expression Analyses

Gene Ontology (GO) enrichment analysis was carried out using Ingenuity Pathway Analysis (IPA, Ingenuity systems, Qiagen, Hilden, Germany), DAVID and STRING 10.5 web based softwares [72,73]. The lists of differentially expressed genes (DEG) after bioinformatics analysis were uploaded into IPA system for core analysis and later overlaid with Ingenuity pathway knowledge base global molecular network. The reported pathways and enrichment analysis was based on −log (*p* values) with a 10% false discovery rate (FDR). Heat maps were generated using a graphical user interface Plotly (https://plot.ly).

### 4.18. qRT-PCR

Isolated total RNA from 1 × 10^5^ cells was reverse transcribed using the Primer Script RT system kit for real time polymerase chain reaction as per manufacturer instructions. The RT-PCR mixture contained SsoAdvanced universal SYBR Green supermix (2×), cDNA template, forward and reverse primers, and nuclease-free water. The RT-PCR was performed on thermocycler (CFX96 Touch, Hercules, CA, USA) using the following cycling parameters: one cycle at 95 °C for 1 min, followed by 40 cycles at 95 °C for 10 s and 60 °C for 60 s. For each experimental sample, a normalized target gene level, corresponding to the relative expression of the target gene with respect to the housekeeping genes (b-actin, Ywhaz, Tbp and Gusb) was determined by 2ΔΔct method, as described previously [74]. 

### 4.19. Western Blot

Whole cell extracts from 5 × 10^5^ cells (6–8 µ-slides) were obtained at 3 h and 6 h post treatment and lysed using cell lysis RIPA buffer (Sigma Aldrich, Schnelldorf, Germany) supplemented with complete mini EDTA free protease cocktail inhibitor (Roche, Germany) and PhosSTOP (Roche, Munich, Germany). The lysates were centrifuged at 10,000× *g* for 30 min at 4 °C. Protein concentrations were determined using Amido black assay and 20 µg of protein in sample buffer was subjected to 7.5–15% SDS-PAGE and transferred onto polyvinylidene difluoride membrane (Immobilon-P, Millipore, Burlington, MA, USA). For immunoblotting, membranes were incubated with primary antibodies as mentioned above (refer to antibodies section). Subsequently, membranes were incubated with horseradish peroxidase conjugated secondary antibodies prior to detection with a lumisensor chemiluminescent HRP substrate using LAS 3000 reader (Fujifilm, Kleve, Germany). Band densities were measured by densitometry (ImageJ software Version 1.51, NIH, Bethesda, MD, USA). Density values are expressed as a ratio normalized to loading control and the fold change is compared to control samples.

### 4.20. Statistical Analysis

All the images were processed using ImageJ (National Institutes of Health, Bethesda, MD, USA), Adobe Photoshop and Inkscape. Data obtained from flow cytometry and ImageJ analyses were calculated with MS Excel (Microsoft Corporation, Redmond, WA, USA) and GraphPad Prism 5 (Version 5.04 Widows, GraphPad, San Diego, CA, USA). Data is presented as mean ± standard error of the mean (S.E.M). Significance amongst groups was conducted via one-way analysis of variance (ANOVA) followed by Bonferroni’s test using Graphpad Prism 5. Significance was accepted at *p* < 0.05, *p* < 0.01, *p* < 0.001: * w.r.t control; $ w.r.t EF; # w.r.t Li.

## Figures and Tables

**Figure 1 ijms-24-03433-f001:**
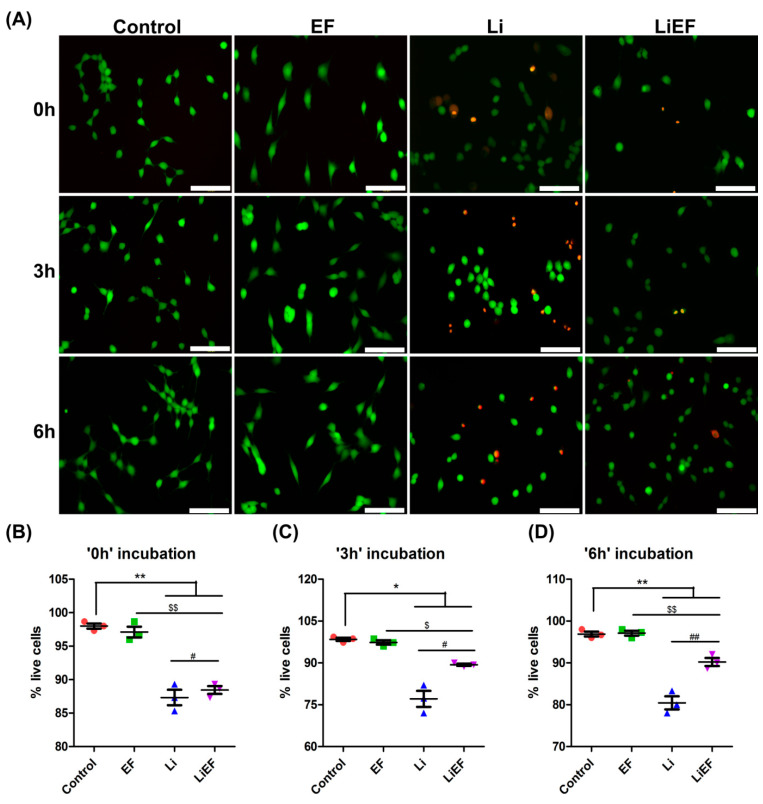
Viability analysis of 661W cells in response to EF stimulation and irradiation. (**A**) Fluorescent images of 661W cells stained with Calcein AM and ethidium homodimer 1 (EthD-1) dyes to quantify live (green) and dead (red) cells respectively at 0 h, 3 h and 6 h time points in: Control, EF, Li, and LiEF. (Scale bar = 100 µm). (**B**–**D**) The graphs represent quantitatively analyzed percent of live cells at 0 h (**B**), 3 h (**C**), and 6 h (**D**) post treatment in Control, EF, Li, and LiEF using ImageJ software (Open source software; National Institutes of Health, Bethesda, MD, USA). (number of cells in triplicates = 150 cells; 3 independent experiments; mean ± S.E.M.; ** *p* < 0.01,* *p* < 0.05 w.r.t Control; $$ *p* < 0.01, $ *p* < 0.05 w.r.t EF; ## *p* < 0.01, # *p* < 0.05 w.r.t Li; One way-ANOVA followed by Bonferroni’s test).

**Figure 2 ijms-24-03433-f002:**
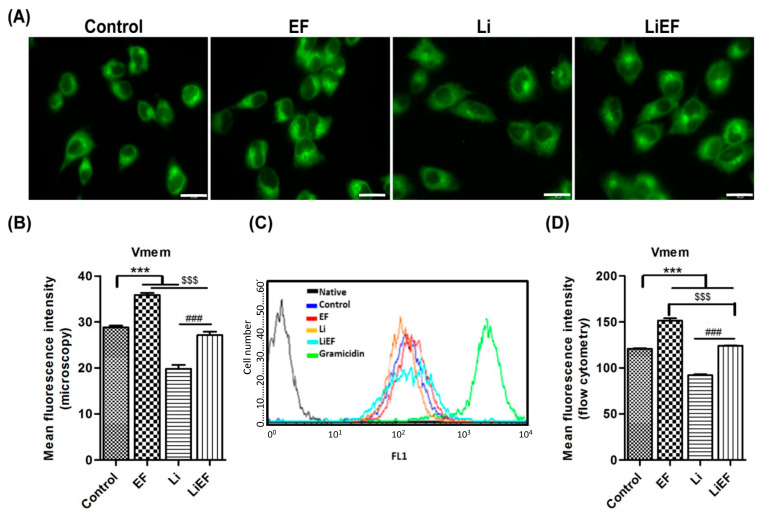
EF depolarizes plasma membrane potential in irradiated 661W cells. EF induced membrane depolarization was determined by an anionic fluorescent dye DiBAC_4_ (3). (**A**) Representative images of the fluorescence intensity of cells imaged using fluorescence microscope in all treated samples at 0 h. Increased fluorescence intensity was observed in EF stimulated cells and a decreased fluorescence intensity in Li-treated cells compared to control cells. (Scale bar = 20 µm). (**B**) Quantitative analysis of fluorescence intensity of the dye in all samples measured at fluorescence microscope. (**C**) Histogram representing mean DiBAC_4_ (3) fluorescence intensity profile obtained from flow cytometry at 0 h of different samples is shown as FL1 on *x*-axis. Gramicidin-treated cells were used as a positive control. (**D**) Quantitative analysis of fluorescence intensity of the dye measured using flow cytometry at different treated conditions: Control, EF, Li, and LiEF (number of cells in triplicates = 40 (microscopy); 5000 (flow cytometry); 3 independent experiments; mean ± S.E.M; *** *p* < 0.001 w.r.t control; $$$ *p* < 0.001 w.r.t EF; ### *p* < 0.001 w.r.t Li; One way-ANOVA followed by Bonferroni’s test).

**Figure 3 ijms-24-03433-f003:**
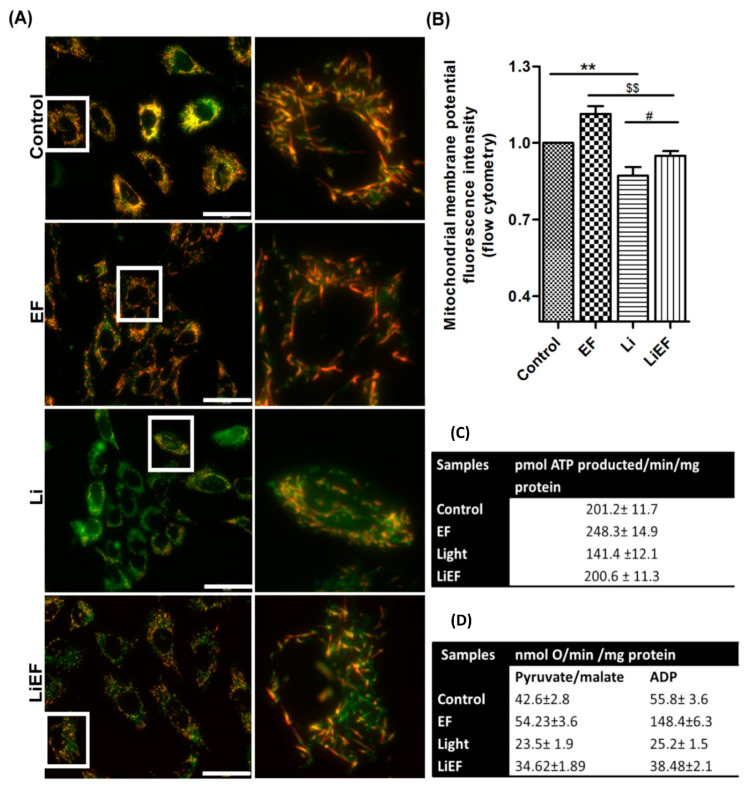
Gain in mitochondrial membrane potential, oxygen consumption and ATP production in irradiated cells due to EF stimulation. (**A**) Representative images of JC-1 dye during different treated conditions at 0 h time point (Scale bar = 50 µm) and enlarged picture of the marked cells. (**B**) Quantified mean fluorescence intensity ratio of aggregate (red color in (**A**)) to monomer (green color in (**A**)) JC-1 form measured using FACS analysis (number of cells in triplicates for flow cytometry = 10,000 cells; 3 independent experiments; mean ± S.E.M.; ** *p* < 0.01 w.r.t Control, $$ *p* < 0.01 w.r.t EF, # *p* < 0.05 w.r.t Li; One way-ANOVA followed by Bonferroni’s test). (**C**) Mitochondrial oxygen consumption was assessed in coupled condition using an amperometric electrode. It reports a representative data of the oxygen consumption at 0 h after addition of 5 mM Pyruvate, 2.5 mM Malate and 0.3 mM ADP to stimulate the process. Irradiated cells displayed decreased mitochondrial respiration, while EF rescued the mitochondrial respiration levels as in control samples. (Number of experiments = 3; mean ± S.E.M.). (**D**) ATP production was assessed using luciferin/luciferase chemiluminescent method at 0 h. In the presence of 5 mM Pyruvate and 2.5 mM Malate as substrates, irradiated cells displayed an inhibition of ATP synthesis by 30%, while EF rescued the ATP synthesis levels as in the control samples. (Number of independent experiments = 3; mean ± S.E.M.).

**Figure 4 ijms-24-03433-f004:**
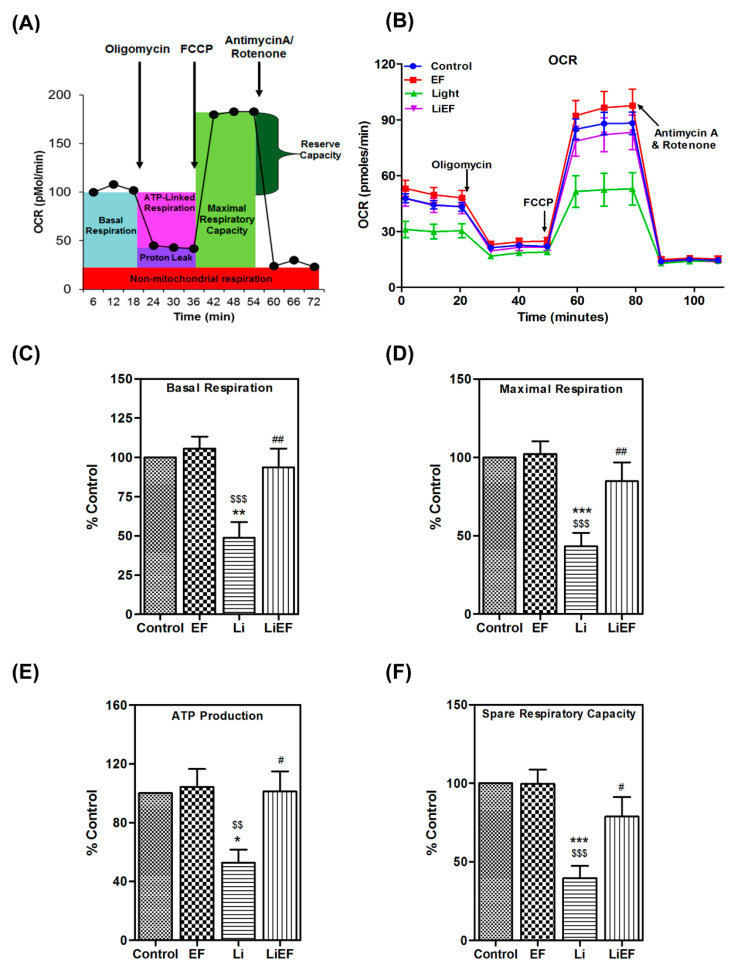
Representative OCR (Oxygen Consumption Rate) bioenergetics profile in 661W cells. (**A**) General schematic representation of OCR bioenergetics profile in relation to time following addition of oligomycin (complex V inhibitor), the uncoupler FCCP (protonophore) and the electron transport inhibitor antimycin/rotenone (inhibitors of complex III/I). (**B**) OCR graph reflects the mitochondrial respiration in 661W cells. The bar graphs represent calculated (**C)** Basal respiration, (**D**) Maximal respiration, (**E**) ATP production, and (**F**) Spare respiratory capacity. (mean ± S.E.M.) (N = 6–10 replicates, 3 independent experiments. Significance was accepted at *** *p* < 0.001, ** *p* < 0.01,* *p* < 0.05 w.r.t Control; $$$ *p* < 0.001, $$ *p* < 0.01, w.r.t EF; ## *p* < 0.01, # *p* < 0.05 w.r.t Li; One way-ANOVA followed by Bonferroni’s test).

**Figure 5 ijms-24-03433-f005:**
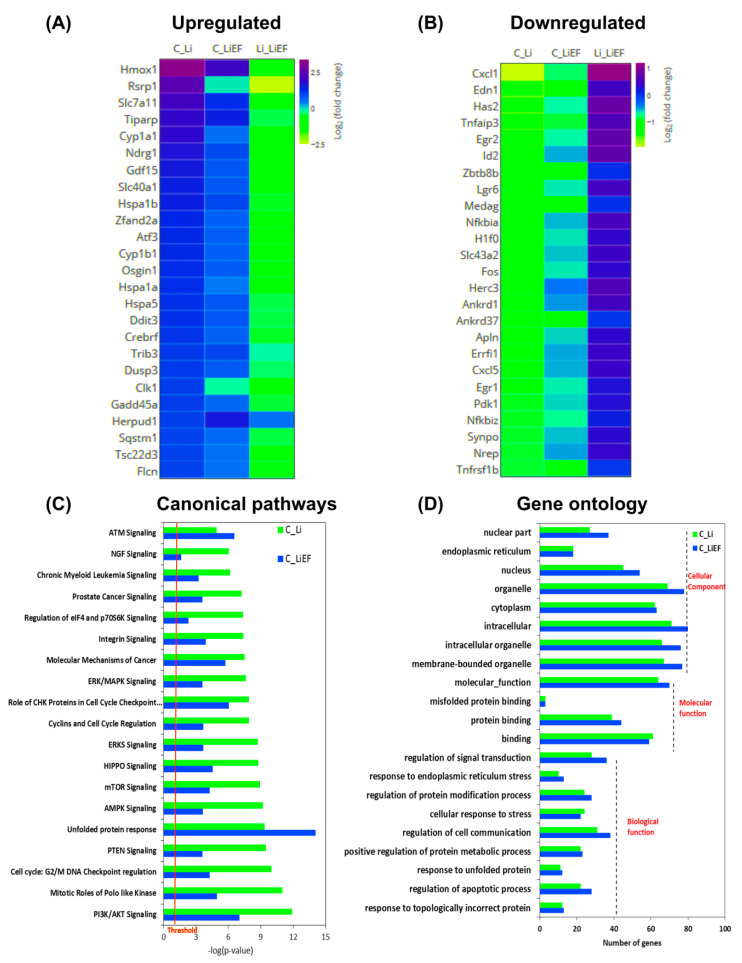
Pathway analysis and Gene ontology (GO) of differentially expressed genes (DEGs). (**A**,**B**) Heat map of top 25 upregulated and downregulated genes based on log_2_(fold change) generated using graphical user interface Plotly. (**C**) Top canonical pathways derived from ingenuity pathway analysis (IPA) software upon core analysis (cut off *p* value of 6). Threshold represents the minimum significance level (scored a −1log(*p*-value) from Fisher’s exact test, set here to 1.25). (**D**) The functional enrichment of top 100 upregulated proteins in the interaction network was carried out in STRING 10.5 database web browser. Red font means the category of cell biological function. Only the top few most significantly enriched GO terms in each GO category (Biological Process, Cellular Component and Molecular function) were presented. Horizontal axis represents the number of genes. A GO term was considered significant at *p*-value < 0.01. (Number of experiments = 3).

**Figure 6 ijms-24-03433-f006:**
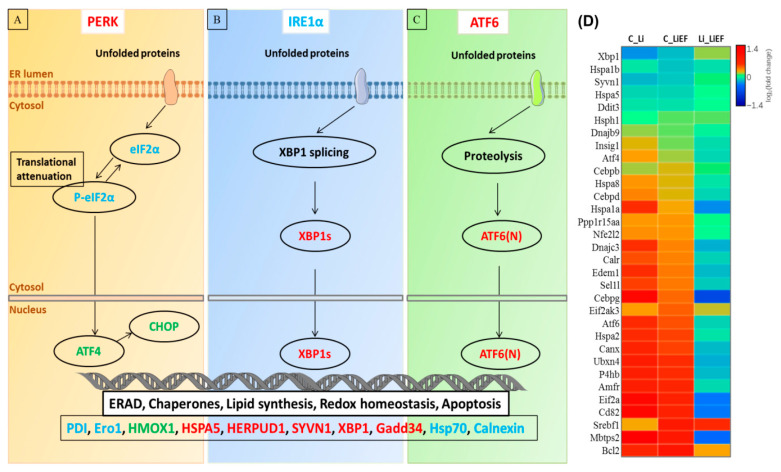
Unfolded protein response (UPR) pathway. (**A**–**C**) Schematic representation of UPR pathway. The UPR pathway is mediated by three different branches: (**A**) PERK-eIF2α-ATF4-CHOP branch, (**B**) IRE1-XBP1 branch, and (**C**) ATF6 branch. These three branches are indicated by different colored font. Markers investigated in this study are highlighted in red (RNA), blue (protein) and green (both RNA and protein). (**D**) Heat map of Differentially Expressed Genes (DEGs) involved in UPR pathway based on log_2_(fold change) obtained from RNA-Sequencing. Heat map is generated using a graphical user interface Plotly.

## Data Availability

Not applicable.

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
