# Peer review of "Analysis of Electric Field Stimulation in Blue Light Stressed 661W Cells"

_ijms, 2023, doi:10.3390/ijms24043433_

Round 1

Reviewer 1 Report

The article entitled "Analysis of electric field stimulation in blue light stressed 661W cells" by Bola et al. reveals that  EF stimulation induced protective effects in 661W cells from Li-induced stress by multiple defense mechanisms such as increase in mitochondrial activity, gain in mitochondrial potential, increase in superoxide levels, and activation of unfolded protein response (UPR) pathways, all leading to an enchanced cell viability and decreased DNA damage

The topic is original or relevant in the field and addresses a specific gap in the field.
The methods are well described, results are organized, clear and well presented. The references appropriate and conclusions are supported by their findings.

I have one question for the authors. They have reported as in Fig S3, that the suppression of mitotic activity of 661W cells exposed to Li is not recovered after 6h of exposure to EF. I wonder if a longer window might be needed to judge this fact., and would like to see more discussion of this point in the discussion area. 

Author Response

Thank you for your valuable comments

We have now explained why we have a methodological restriction to prolong the cell observation experiments longer than six hours. At 12 or 24 hours, the culture is becoming confluent and so an exact analysis was not possible anymore.

See also letter to the editors

Reviewer 2 Report

Electrical stimulation is a method used therapeutically to treat retinal and spinal cord injuries. This study is very valuable in terms of elucidating the protective mechanism at the cellular level. In this study, it is seen that the immediate effects of cellular events such as viability, oxidative damage, DNA damage and cell migration were examined and mitochondrial activity measurements were made in blue light (Li) stressed 661W cells subjected to direct current electric field stimulation. It is also stated that the UPR pathway involved in alleviating Li-induced stress has been studied.

The study was well designed and presents a methodology compatible with the results and conclusions, with interesting results that deserve to be published. The topic is important and the authors collected comprehensive literatures in the field. I would suggest that the manuscript can be considered for publication if the authors can improve the manuscript in following aspects:

However, one of the main limitations of the study, from the point of view of this reviewer, which should be pointed out in the introduction, is the aim is not expressed in an intelligible way according to hypothesis.

The introduction section was incomplete in terms of presenting the background of the study. In the introduction section, findings are given in line with the hypothesis. In fact, it would be much more appropriate if this information and data were explained in the discussion section. In addition, the introduction part of this valuable study can be better emphasized by explaining the subject and purpose of the study instead of giving findings.

Current references should be included in addition to the references cited.

Author Response

Thank you for yours valuable comments.

We have now changed the Introduction and put a large part of the protective mechanisms of electrical stimulation into Discussion. On the other hand, we have emphasized the problem of blue light hazard, which is increasingly present in our environment - be it ambient illumination, or illumination of electronic screens. Thus, the aim of our study is to go deeper into protective mechanisms of retinal photoreceptor cells via electrical stimulation (see Introduction). This electrical stimulation is already used in clinical settings, however, the knowledge of the molecular mechanisms in retinal cells is incomplete.

Furthermore, we have added recent literature references to both, blue light hazard and protective mechanisms by electrical stimulation.

All changes are depicted in red.

Round 2

Reviewer 1 Report

I recommend this article to be accepted for publication 

Reviewer 2 Report

The corrections suggested by the reviewers were made correctly.

It is appropriate to publish the manuscript.